# SubvectorS_Geo: A Neural-Network-Based IPv6 Geolocation Algorithm

**Zhaorui Ma** [1,2,3,†], **Xinhao Hu** [2,*,†], **Shicheng Zhang** [2], **Na Li** [1,4], **Fenlin Liu** [1], **Qinglei Zhou** [1], **Hongjian Wang** [2], **Guangwu Hu** [3,*] and **Qilin Dong** [2]

1 State Key Laboratory of Mathematical Engineering and Advanced Computing, Zhengzhou 450052, China
2 School of Computer and Communication Engineering, Zhengzhou University of Light Industry, Zhengzhou 450002, China
3 Shenzhen Institute of Information Technology, School of Computer Sciences, Shenzhen 518172, China
4 School of Electronic Information Engineering, Sias University, Zhengzhou 451150, China
* Correspondence: huxinhao@email.zzuli.edu.cn (X.H.); hugw@sziit.edu.cn (G.H.)
† These authors contributed equally to this work.

**Abstract:** IPv6 geolocation is necessary for many location-based Internet services. However, the accuracy of the current IPv6 geolocation methods including machine-learning-based or deep-learning-based location algorithms are unsatisfactory for users. Strong geographic correlation is observed for measurement path features close to the target IP, so previous methods focused more on stable paths in the vicinity of the probe. Based on this, this paper proposes a new IPv6 geolocation algorithm, SubvectorS_Geo, which is mainly divided into three steps: firstly, it filters geographically relevant routing feature codes layer by layer to approximate the fine-grained trusted region of the target; secondly, it extracts delay vectors into the trusted region; thirdly, it evaluates the vector similarity to determine the final target geolocation information. The final experiments show that the median error distance range is 7.025 km to 9.709 km on three real datasets (Shanghai, New York State, and Tokyo). Compared with the advanced method, the median distance error distance is reduced by at least 6.8% and the average error distance is reduced by at least 9.2%.

**Keywords:** IPv6 geolocation; network mapping; neural network



## 1. Introduction

As we know, IP addresses have a dual semantic meaning, representing both identity and location [1]. IP geolocation is a technology that aims to provide high-precision geographic information, such as the country, region, latitude, longitude, and time zone of the host [2]. Actually, IP geolocation is important to Internet users. It has accommodated many location-based services, personalized recommendations, Whois, DNS, public databases, etc. Furthermore, it is an important tool for back-tracing and tracking in the field of network security [3].

Existing IP geolocation algorithms are mainly divided into two categories: (1) database-query-based IP geolocation algorithms, as early research focused on the records provided by public databases including WHOIS and DNS, but the database is poorly updated and maintained, making its geolocation performance degraded; existing research is focusing more on improving geographic position performance by enhancing database data reliability [4]; (2) the network-measurement-based IP geolocation algorithm collects the network measurement data between the probe and the target IP, constructs the measurement data geographic distance mapping with different methods, generates distance constraints for the target IP, and estimates the geographic location information of the target IP [5]; in addition, it can be subdivided into rule-based IP geolocation algorithms and learning-based IP geolocation algorithms according to the different methods; thanks to the real-time and

realistic nature of network measurement data, network-measurement-based methods have gradually become the main research direction in the field of IP geolocation research [6].

The rapid development of IPv6 networks has seen a rise in demand for location-based Internet applications; the previous IPv4 location algorithms did not achieve the same performance in IPv6. For researchers, the design of IP geolocation algorithms is based on IP protocols and the rules of network environments where IP protocols are deployed, and the design of IPv6 is proposed using IPv4 as a blueprint. There are differences between them, but at the same time, there are also similarities, so there is no split between IPv4 location algorithms and IPv6 location algorithms, just because of the differences in the IP protocols and network environment rules they face. Compared with IPv4, IPv6 has changed differently in terms of address pools, access methods, protocols, traffic characteristics, and AS-domain-level paths [7]. In a study related to the traditional IPv4 geolocation algorithm, IPv6-CBG [8] shows that the geolocation error of IPv6 networks is about 30% higher than that of IPv4 in the same experimental region due to the high latency in IPv6 networks; the RNBG [9] algorithm relies on the delay distribution rules in IPv4 and the difference in the delay distribution caused by the topological changes in IPv6. The Corr-SLG [10] algorithm enhances the SLG algorithm in weakly connected networks by evaluating the positive and negative correlations of hosts' relative delay distances, but it cannot improve the localization accuracy for weakly connected hosts and degraded localization performance. In addition, Yang [11] proposed that the traditional IPv4-based network measurement method is affected by the huge address pool of IPv6, and the measurement period becomes longer. In IPv6, some of the traditional IP geolocation algorithms are applied with increased time cost, storage cost, and geolocation errors, so we try to design high-performance IP geolocation algorithms under IPv6.

Deep learning plays an important role in the field of IP localization based on network measurements. For IPv6, we introduce deep learning based on a priori knowledge of IP geolocation. There are three main challenges: (1) a lack of a reasonable evaluation of the true physical distance of the target IP; (2) how to reduce the negative impact of weak connectivity, high latency, and circuitous routing on measurement-based IP geolocation in IPv6 networks; (3) how to interpret the IPv6 geolocation model under deep learning. To address these issues, we designed an IPv6 geolocation algorithm called SubvectorS_Geo, which improves the performance of IP geolocation algorithms, under the assumption of similarity in host latency in the same region. SubvectorS_Geo is first based on a priori knowledge of IPv6 geolocation, using a rule-based approach combined with SubvectorS_Geo. It first constructs a "layer by layer approximation" region constraint based on the a priori knowledge of IPv6 geolocation and, then, uses a rule-based approach to combine landmark measurement data to construct a "layer by layer approximation" region constraint. Within the trusted region, SubvectorS_Geo evaluates the delay similarity from the delay sequences near the target IP, effectively reducing the possibility that delayed similar hosts are not in the same geographic location. In summary, the main contributions of this work are as follows:

- We propose a new IPv6 geolocation algorithm, which solves the problem of the low accuracy of existing geolocation and the lack of reasonable and effective constraints on regional delay similarity.
- We apply residual paths (measured paths in trusted regions) to IPv6 geolocation models, and residual path features have a strong geographic correlation with the target IP. A region constraint strategy is added based on IPv6 prefix similarity to improve the fine-grained trusted region constraint scheme, and IPv6 prefix similarity has a high geographic correlation. To the best of our knowledge, we are the first to introduce residual paths and IPv6 prefix similarity in the IPv6 geolocation domain.
- The final experimental results of our method show that our method outperforms current IPv6 geolocation algorithms in IPv6 geolocation tasks under noncollaborative conditions.

The rest of the paper is organized as follows: Section 2 introduces and analyzes existing IP geolocation algorithms. Section 3 introduces the basic principle and main steps of the hierarchical clustering IPv6 geolocation algorithm, SubvectorS_Geo, based on network

features. Section 4 analyzes and discusses the performance of SubvectorS_Geo and existing IP geolocation algorithms by comparing experimental results. Finally, Section 5 conclude this paper.

The source code of this model has been open sourced to Github https://github.com/Hxh1863819/SubvectorS_Geo/ (accessed on 4 January 2023).

## 2. Related Work

The related work is divided into two parts. The first part unfolds the research content related to IP geolocation, and the second part is the analysis and discussion of several state-of-the-art rule-based and learning-based IP geolocation algorithms. Its IP geolocation idea is shown in Figure 1 following.

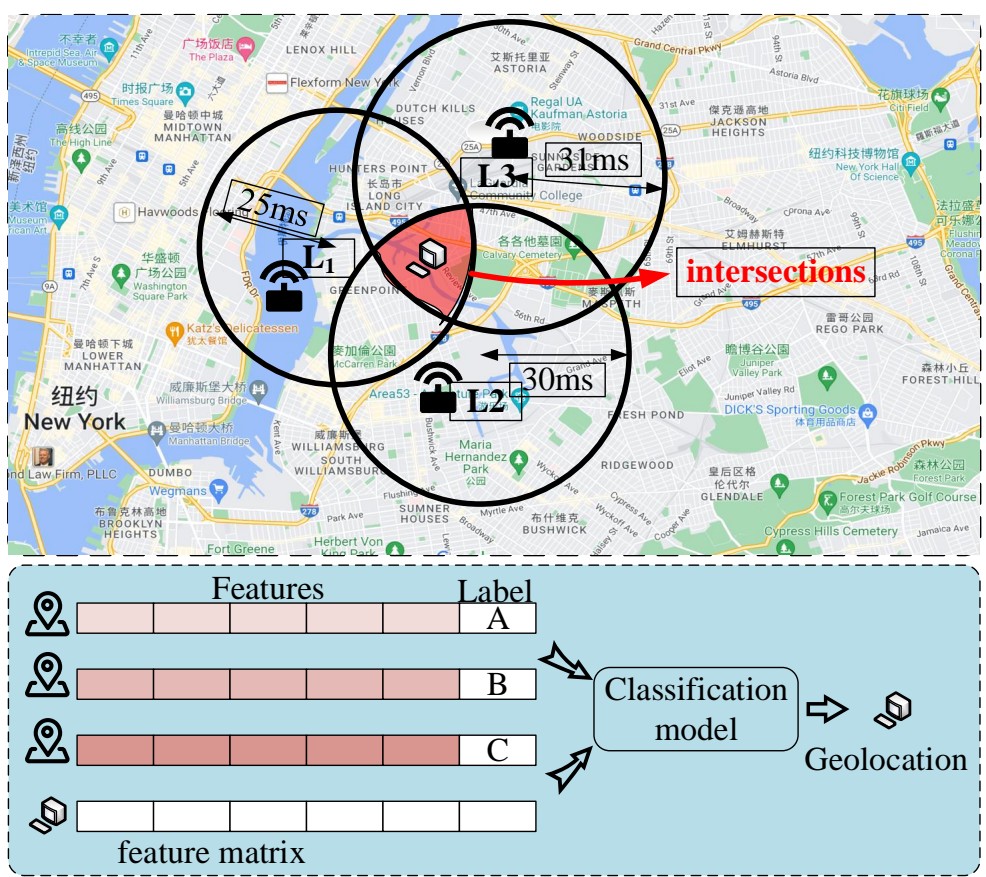

**Figure 1.** IP geolocation algorithm analysis.

### 2.1. IP Geolocation Algorithm

IP geolocation techniques fall into two main categories: database-based queries [12] and network-based measurements [13].

IP geolocation based on database query is realized by querying the corresponding location information of the target IP recorded in the IP geolocation database [14]. The IP geolocation database records and maintains a series of geographic information records of IP addresses [4]. The IP geolocation database query method is faster, but its data reliability cannot be verified [15]. On the other hand, data integrity and real-time performance are poor because data resources are updated slowly [16]. Contemporary researchers more often use multidatabase federated queries and learning-based methods to improve their geolocation performance [17] and databases [18]. Although having different degrees of accuracy improvement, they still cannot be used to accomplish fine-grained IP geolocation tasks.

The network-measurement-based IP geolocation method obtains the data [19] of Internet host network communication using active or passive network measurement methods.



The measurement data are analyzed and processed to establish the mapping relationship with the geographic location to achieve IP geolocation. According to the differences in research methods that are divided into two categories: (1) There are the rule-based IP geolocation algorithms such as SLG [20], Corr-SLG [10], RNBG [9], and Geo-PoP [21]; network characteristics combined with the analysis of the real geographic location are used to discover the corresponding rules. The core rule of SLG is to filter the landmarks with the smallest relative delay to the target IP, based on the assumption that the relative delay size is proportional to the distance. However, this rule has poor performance in weak connections. RNBG is based on the assumption that the delay size is proportional to the real physical distance. The core rule is to screen the largest one-hop delay on the path and use the neighboring router with the largest delay as the city-level boundary route, which circumvents the impact of network connectivity on IP geolocation accuracy, but cannot complete the fine-grained IP geolocation task. Rule-based IP geolocation algorithms have good performance in the networks to which their rules apply, but the rules cannot be applied to all networks, Therefore, the generalization ability of this type of algorithm is poor. (2) The learning-based IP geolocation algorithms include LBG [22], TNN [23], MLP-Geo [24], GeoCET [25], etc. LBG creatively introduces regional demographic factor features and uses machine learning methods to statistically analyze network measurement data to complete city-level IP geolocation tasks. GeoCET completes IP geolocation tasks by combining elliptical trajectory constraints with machine learning techniques. Learning-based IP geolocation algorithms can better improve the generalization ability of IP geolocation algorithms. However, we need to pay attention to the interpretability of IP geolocation models in deep learning.

As more local servers are deployed in cloud-based solutions, the number of landmarks available for collection becomes smaller, posing new challenges for measurement-based IP geolocation [10]. The combination of IPv6 and the Internet of Things (IoT) [26] has given birth to many new IPv6 application scenarios [27]. There are also studies on IP geolocation algorithms based on the application scenarios of IoT devices. For example, GeoCAM [28] select cameras as landmarks in IoT application scenarios to design fine-grained IP geolocation algorithms. Ding [10] proposed a new landmark collection technique that uses mobile devices to collect WiFi [29] landmarks in public places such as hotels, banks, and supermarkets. Rye [30] proposed an IPv6 geolocation algorithm based on home routing devices, which first collects MAC addresses that are embedded in IPv6 addresses and, then, queries the target IP geolocation by WiFi BSSIDs [31]. The algorithm has high accuracy; however, not all MAC addresses are embedded in IPv6 addresses, and the algorithm has poor generalization ability. The robustness of geographic information from detectable IoT devices meets the requirements of landmarks while providing a finer granularity of geographic information.

### 2.2. Analysis of Existing Advanced IP Geolocation Algorithms

The existing IP geolocation idea is shown in Figure 1. IPv6-CBG [8] is an early IPv6-based geolocation study, and the algorithm improves the accuracy of IPv6 geolocation algorithms by constructing a delay distance mapping function to evaluate IPv6 geolocation tasks. Although the algorithm is an IPv6 geolocation study based on the delay value, the delay value itself is affected by the network congestion, which makes it impossible to represent the real physical distance. In addition, the high delays in IPv6 environments increase the actual geolocation errors of the algorithm. Corr-SLG [10] is an advanced rule-based IP geolocation algorithm that evaluates the positive and negative strong correlations of landmarks relative to the delay distance and classifies landmarks into positive strong correlation, negative strong correlation, and weak correlation, enhancing the application of SLG in weakly connected networks at the expanse of hosts in the weak correlation class unable to be accurately located. The performance of the rule-based approach relies on host partitioning, which follows a preassumed linear rule. In weakly connected IPv6,

the presence of a large number of hosts in the weakly correlated class increases the true geolocation error of the algorithm.

The TNN algorithm [23] and the MLP-Geo algorithm [24] give up observing the relationship between delay values and physical distances. The TNN algorithm evaluates the geolocation information of the target IP by training the neural network to learn the mapping function of delay similarity markers, which has excellent geolocation performance. However, dynamic updating routes can affect the evaluation of delayed similarity. There exist delay similar hosts with a real physical distance that is far, ignoring the need for regional delay similarity for fine-grained reasonable region constraints. The MLP-Geo algorithm adds a region constraint to the region delay similarity assumption by adding stable paths, which makes up for the lack of the region constraint in the TNN algorithm and enhances the application of region delay similarity in IP geolocation algorithms. However, it is worth noting that the stable path is a set of paths composed of stable routing device connections, which can effectively represent the coarse granularity of packet traces in the network. However, the stable path tends to focus on the side of the measurement path near the probe, and there are fewer stable routes on the side near the target IP. The stable path ignores the features of the measurement path near the target host and, likewise, cannot constrain the path after the stable path direction. In other words, the stable path cannot constrain the fine-grained region and path direction of the target IP.

Delay similarity [23] helps us in the task of geolocating IPv6 hosts. Since different regions in the real world may also have similar host delay, delay similarity cannot be a decisive factor, and it requires a fine-grained geographic constraint. The difference between our work and existing methods is that we focus on the regions that cannot be covered by stable paths, extract geographically constrained routing features hierarchically from landmark measurement paths, approximate the target IP to a fine-grained trusted region, and focus on host delay similarity within this region for accurate and robust IPv6 geolocation.

## 3. The SubvectorS_Geo Algorithm

The algorithm is implemented through four modules: the preprocessor module, the encoder module, the preclassifier module, and the neural network module. The preprocessor module is used to collect and collate network measurement data. The encoder module is used to encode information about network edge path features close to the target IP. The preclassifier module performs street-level landmark clustering to reduce model complexity. The neural network module learns delayed similarity in large-scale data.

### 3.1. Algorithm Overview

Through a large amount of measurement data, the hypothesis that hosts in the same region have a similar delay distribution and similar IP address prefixes, while the closest common router near the edge end of the network on the path can determine the path direction can be verified. Based on this fact, the following hypothesis is proposed: there exists a similar geographical region with similar delay vectors and similar IP address prefixes for hosts in a fine-grained trusted region. The design is based on the idea of layer by layer approximation, and a coarse-grained–fine-grained target IP trusted region is delineated after rule filtering of AS domain names, identification IPs, closest common routers, common prefixes of terminal addresses, and the closest routes with similar hostnames in the order of the measurement paths. The existence of the closest common router specifies the consistency of path directions. The common prefixes of terminal addresses characterize the target IP near path address prefixes that characterize the paths, while the IPv6 address segment in the Whois information verifies the accuracy of the common prefixes. Finally, the target IP geolocation information is determined within a fine-grained trusted region with the result of regional delay similarity evaluation, and the task of IPv6-based IP geolocation is finally completed. Based on this assumption, this paper proposes a new IPv6 geolocation algorithm framework, as shown in Figure 2.

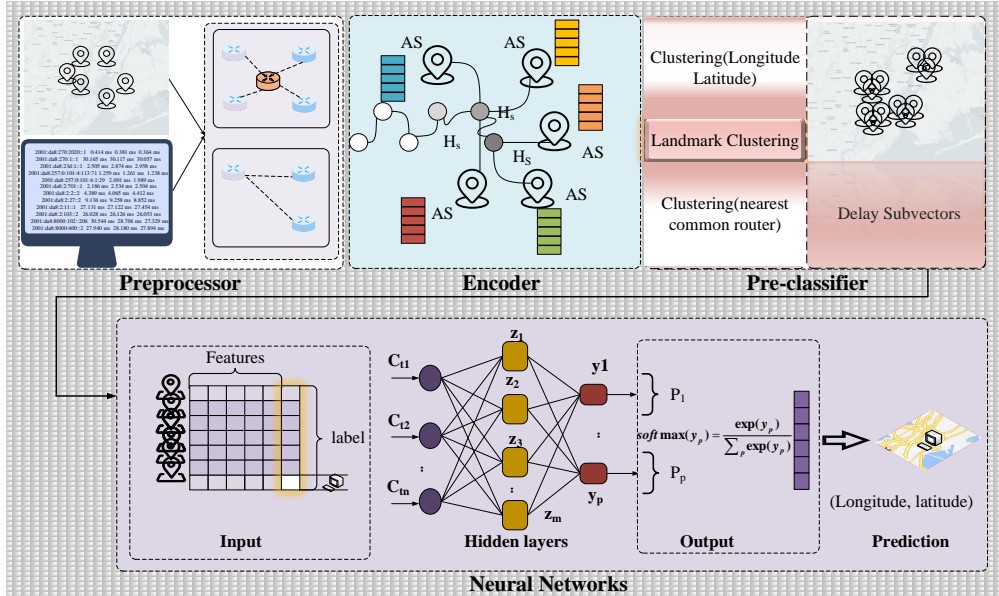

**Figure 2.** Overview of the proposed IP geolocation framework.

### *3.2. Preprocessor*

The measurements are derived by probing landmark traceroute measurements, and the main task of the preprocessor section is data collection and cleaning. Task 2 is the encoding task of the raw measurement data.

Traceroute: Suppose there are N landmarks in the network, and according to the M probes we deploy, traceroute measurements are sent from the probes to each landmark to obtain a set of M*N landmark path measurement datasets, with the geographic information of the landmarks in the form of (longitude, latitude) as labels. It is worth noting that this paper focuses on delay similarity. Therefore, we added the measurement cycle to collect stability delay information.

Aliases and anonymous routers: The presence of alias routes and anonymous routers also significantly affects the model's focus on routing vectors. For an anonymous router, we tried to eliminate it directly and only retained its connectivity in the network topology. For the alias router, because it leads to a diversity of measured paths, we focused on the path segment in which the alias route is located to discriminate similarly with other routing environments and evaluate whether the alias route is eliminated.

### *3.3. Encoder*

Route-hop code method: Take the delay features and IP address features as the basis for IP geolocation.

### 3.3.1. Dataset Construction

Landmark: The landmark is the collated measurement data. M probes are sent to N landmarks to construct the M*N measurement path dataset $E_{ld}$ and the delay dataset $T_{ld}$ corresponding to the hop value, combined with the Rapid7 query to build the landmark measurement path hostname $H_{ld}$. The hostname information is set to zero if not queried, and the landmark router path set $E_{ld}$ is analyzed to collect the identity IP set $R_i$.

$$E_{ld} = (p_{m1}, p_{m2}, p_{m3}, ..., p_{mn}) \tag{1}$$

$$T_{ld} = (t_{m1}, t_{m2}, t_{m3}, ..., t_{mn}) \tag{2}$$

$$H_{ld} = (h_{m1}, h_{m2}, h_{m3}, ..., h_{mn}) \tag{3}$$

$$R_i = (r_{1i}, r_{2i}, r_{3i}, ..., r_{zi}) \tag{4}$$

$p_{mn}, t_{mn}$, and $h_{mn}$ record a complete path from the $m$th probe to the $n$th landmark router information, delay information, and hostname information. It is worth noting that the data here are stored in the order of path hop values to facilitate subsequent data filtering work. $r_{zi}$ is the identification IP collected for the experimental region according to Zhao's method [13].

Target IP: Data probing with M probes for X target IPs to be located. The obtained raw measurement data are cleaned to obtain the target IP path set $E_{td}$ and the delay dataset $T_{td}$. Build the landmark measurement path hostname dataset $H_{td}$ in combination with the Rapid7 query, and set to zero if the hostname information is not queried.

$$E_{td} = (p_{m1}, p_{m2}, p_{m3}, ..., p_{mx}) \tag{5}$$

$$T_{td} = (t_{m1}, t_{m2}, t_{m3}, ..., t_{mx}) \tag{6}$$

$$H_{td} = (h_{m1}, h_{m2}, h_{m3}, ..., h_{mx}) \tag{7}$$

$p_{mx}, t_{mx}$, and $h_{mx}$ record a complete path from the $m$th probe to the $x$th target router information, delay information, and hostname information.

### 3.3.2. Closest Common Router Set

Construction of the closest common routers set: Combined with the identity IP set $R_i$, all routing addresses in each path due to the identity IP (in the direction of the path hop value) are filtered from the landmark path dataset $E_{ld}$ as the closest common router set $E_{lrp}$. Its corresponding hostname dataset is also filtered out in the order of hop value within the hostname dataset of the closest common router $H_{lrp}$.

$$E_{lrp} = (R_{nc(m1)}, R_{nc(m2)}, R_{nc(m3)}, ..., R_{nc(mn)}) \tag{8}$$

$$H_{lrp} = (h_{nc(m1)}, h_{nc(m2)}, h_{nc(m3)}, ..., h_{nc(mn)}) \tag{9}$$

$R_{nc(mn)}$ and $h_{nc(mn)}$ record the complete path due to the identification IP, the $m$th probe to the $n$th landmark probe path router information, and hostname information.

### 3.3.3. Path Encoding

Landmark path encoding and entity encoding to be measured in $E_{ld}$. The feature encoding is performed for M*N landmark paths.

$$C_l = (R_{cl}, T_l) \tag{10}$$

$$C_l = (AS, P_i, L, P_{mn(/\alpha)}, H_s, T_l) \tag{11}$$

$$C_t = (R_{ct}, T_t) \tag{12}$$

$$C_t = (AS, P_i, L, P_{mx(/\alpha)}, H_s, T_t) \tag{13}$$

$C_l$ and $C_t$ are the path coding vectors of landmarks and targets. $R_{cl}$ and $R_{ct}$ are the route feature coding vectors of landmarks and targets. The dimensions of the $R_{cl}$ and $R_{ct}$ vectors and the encoded features are the same: $AS$ is the AS domain information code of the terminal IP. $P_i$ is the encoded value that identifies the IP. $L$ is the value of the code that evaluates the presence of the closest common router. $P_{mn(/\alpha)}$ is the common address prefix ofthe landmark and the closest common router. $P_{mx(/\alpha)}$ is the common address prefix of the target and the closest common router. $H_s$ is the encoded value to evaluate the presence of similar routes for hostnames. $T_l$ and $T_t$ are the remaining path delay vector encoding of the network for the landmark and the target in the trusted region approximated by the routing feature, respectively.

Routing vector encoding: Coding tasks for geographically correlated routing features in the traceroute measurement data.

$R_{cl}$ and $R_{ct}$ are encoded representations of the routing features for landmark paths and destination paths. The coding features and coding methods are the same. The detailed process for routing code ($R_{cl}$) for landmark paths is as follows.

$$P_i = \begin{cases} P_{m,n,i} & \text{if } P_{m,n,i} \text{ in } R_i \\ \gamma & \text{if } P_{m,n,i} \text{ not in } R_i \end{cases} \quad (1 < i < k) \tag{14}$$

$$L = \begin{cases} 1 & \text{if } P_{m,n,j} \text{ in } E_{lrp} \\ 0 & \text{if } P_{m,n,j} \text{ not in } E_{lrp} \end{cases} \tag{15}$$

$P_{m,n,k}$ is the *k*th IPv6 routing address on the probing path of the *m*th probe in the $E_{ld}$ dataset for the *n*th landmark route or the terminal routing address. In the screening evaluation of identification IPs, to pursue the geographically strong correlation of identification IPs for terminal routing, the inverse path order for screening is adopted. That is, the order starting from the terminal IP($P_{m,n,k}$) to the end of the starting IP($P_{m,n,1}$). When $P_{m,n,i}(1 < i < k)$ exists in the identity IP set $R_i$, $P_i$ takes the value of the identity IP route $P_{m,n,i}$. Otherwise, $P_i$ takes the value of $\gamma$.

In the screening task of the closest common router: $P_{m,n,j}$ is the any route IP in a path after the identification IP. The path segment starting from the identification IP($P_{m,n,i}(1 < i < k)$) to the terminal route IP($P_{m,n,k}$) screens the closest common router near the terminal in the reverse path order. When any IPv6 route ($P_{m,n,j}(i < j < k)$) on the path exists in the closest common router set $E_{lrp}$, $L$ is assigned a value of 1. Else, $L$ is assigned a value of 0.

$$P_{mn(/\alpha)} = \begin{cases} P_{mn(/\alpha)} & \text{if } P_{m,n,j(/\beta)} = P_{m,n,k(/\beta)} \text{ and } \beta > \alpha \\ \lambda & \text{else} \end{cases} \tag{16}$$

$P_{mn(/\alpha)}$ approximates the trusted region based on the closest common router. It likewise supports the credibility of the region delineated by the closest common router. $P_{mn(/\alpha)}$ analyzes the geographic association of the closest public router and terminal IPs by filtering their common prefixes, taking the value of $P_{mn(/\alpha)}$ when the number of common prefix bits of the closest common router and terminal IP satisfies $\beta > \alpha$. Otherwise, $P_{mn(/\alpha)}$ takes the value of $\lambda$. The $\alpha$ assignment takes into account the address prefix segment provided in the Whois information as a way to limit the common prefix.

$$H_s = \begin{cases} 1 & \text{if } h_{m,n,c} \text{ similar } h_{nc} \ (h_{nc} \text{ in } H_{lrp}) \\ 0 & \text{else} \end{cases} \tag{17}$$

$H_s$ is our filtering evaluation for the hostname similar routers after the closest common router in the landmark measurement data, the path segment starting from the closest common router $P_{m,n,j}$ to the terminal routing address $P_{m,n,k}$. When any IPv6 route $P_{m,n,c(j<c<k)}$ on the path segment has a hostname similar to the one in the $H_{lrp}$ hostname data, i.e., there is a route with a similar hostname after the closest common router, the value is assigned as 1. Otherwise, the value is assigned as 0.

Delay vector encoding: The different trusted regions divided by different routing features determine the different delay vectors of interest, as shown below.

$$T_l = \begin{cases} t_{m,n,k}, t_{m,n,(k-1)}, \ldots, t_{m,n,(i+1)}, t_{m,n,i} & \text{if } P_i = P_{m,n,i} \text{ and } L = 1 \\ t_{m,n,k}, t_{m,n,(k-1)}, \ldots, t_{m,n,j}, \ldots, t_{m,n,1} & \text{if } P_i = \gamma \text{ and } L = 1 \\ t_{m,n,k}, t_{m,n,(k-1)}, \ldots, t_{m,n,j}, \ldots, t_{m,n,1} & \text{if } P_i = \gamma \text{ and } L = 0 \end{cases} \tag{18}$$

$T_l$ landmark delay vector coding: In the landmark path delay dataset $T_{ld}$, when $P_i = P_{m,n,i}$ and $L = 1$, the path coding delay vector $T_l$ is assigned to all delays from the terminal IP delay $t_{m,n,k}$ to the marking IP delay $t_{m,n,i}$, and the inverse path order delay is assigned. When $P_i = \gamma$ and $L = 1$, the assignment of this path delay vector code $T_l$ starts from the terminal IP delay $t_{m,n,k}$ to the end of the starting router delay $t_{m,n,1}$, also called the inverse path order delay assignment. When $P_i = \gamma$ and $L = 0$, the assignment of this path

delay vector code starts from the terminal IP delay $t_{m,n,k}$ to the end of the starting router delay $t_{m,n,1}$, also called the inverse path order delay assignment.

$$T_t = \begin{cases} t_{m,x,k}, t_{m,x,(k-1)}, \cdots, t_{m,x,(j+1)}, t_{m,x,j} & \text{if } P_i = P_{m,x,i} \text{ and } L = 1 \\ t_{m,x,k}, t_{m,x,(k-1)}, \cdots, t_{m,x,(i+1)}, t_{m,x,i} & \text{if } P_i = P_{m,x,i} \text{ and } L = 0 \\ t_{m,x,k}, t_{m,x,(k-1)}, \cdots, t_{m,x,(j+1)}, t_{m,x,j} & \text{if } P_i = \gamma \text{ and } L = 1 \\ t_{m,x,k}, t_{m,x,(k-1)}, \cdots, t_{m,x,j}, \cdots, t_{m,x,1} & \text{if } P_i = \gamma \text{ and } L = 0 \end{cases} \tag{19}$$

$T_t$ target delay vector encoding: In the target path delay dataset $T_{td}$, when $P_i = P_{m,x,i}$ and $L = 1$, the path encoding delay vector $T_t$ is assigned to all the delays from the terminal IP delay $t_{m,x,k}$ to the closest common route IP delay $t_{m,x,j}$, in the reverse path delay order. When $P_i = P_{m,x,i}$ and $L = 0$, the path encoding delay vector $T_t$ is assigned to all the delays from the terminal IP delay $t_{m,x,k}$ to the identified IP delay $t_{m,x,i}$. When $P_i = \gamma$ and $L = 1$, the path encoding delay vector $T_t$ is assigned to all the delays from the terminal IP delay $t_{m,x,k}$ to the closest common router delay $t_{m,x,j}$, also called the inverse path delay order. When $P_i = \gamma$ and $L = 0$, this path delay vector code $T_t$ is assigned from the terminal IP delay $t_{m,x,k}$ to the starting route $t_{m,x,1}$, also called the inverse path delay order. $P_{m,x,i}$ is the representation of the identification IP in the target IP path, which is encoded in the same way as $P_{m,n,i}$.

### 3.4. Pre-Classifier

A robust model requires a reasonable set of parameters. In the assumption of region delay similarity, we must approximate the training set to a fine-grained region to achieve the task of robust model training. Reasonable constraints should be imposed on the partitioning of the training set before, importing the partitioned training set into the neural network model.

Landmark clustering: This task aims to reduce the complexity of the model in evaluating vector similarity through rule-based preselection of real geographic relationships among landmarks. The street-level clustering task of landmarks is accomplished based on a priori knowledge of street-level IP geolocation combined with the routing features of landmark IPs. In addition, as mentioned in the Introduction, the latitude of delay caused by the processing of delay in this paper is not uniform, which leads to the accuracy of the subsequent evaluation results being affected. In the landmark clustering task, we try to solve this problem using SubvectorS.Detailed steps are shown below.

Step 1: First, the closest common router with the strongest and stable geographic correlation is extracted from the coded landmark routing vector as the first round of screening criteria, and then, the IPv6 prefix in the Whois information is used as a constraint to screen for the dataset $G_i(i = 1, 2, 3, \ldots, i)$. $i$ is the maximum profile factor.

Step 2: Clustering based on real geographic information of landmarks (latitude and longitude) to obtain datasets $L_j(j = 1, 2, 3, \ldots, j)$, where $j$ is the maximum profile factor. The clustering method uses the K-means method with a latitude and longitude distance constraint of 10 km.

Step 3: Street-level landmark dataset $G_i \cap L_j$. Obtain $s_n(n = 1, 2, 3, \ldots, n)$, where $n$ is the maximum profile factor and uses the cluster center landmark as the geographic location information for this street-level landmark class.

Step 4: Landmark path collection $S_n(n = 1, 2, 3, \ldots, n)$; the $S_1$ set of geographically related landmarks' path sets (10 km) is recorded, and this section iterates the delay vector on the path of each landmark using the subvector algorithm to obtain the delay vector of a landmark at different latitudes.

SubvectorS: To delineate the final fine-grained region constraint of the entity, we used the rules of the closest common router and the closest hostname similar router. Each entity to be located may filter out different closest common routers and closest routers with similar hostnames. Therefore, after the credible region of each target to be located is divided, its remaining path length is difficult to unify, resulting in the nonuniformity of the

dimensionality of the delay feature vector. Difficulties exist in the similarity measure of the delay vectors based on the cosine similarity.

We let $T_l$ denote the landmark residual path delay vector. $T_t$ denotes the target IP residual path delay vector. $|T_t|$ denotes the dimension of the target IP residual path delay vector. SubvectorS enumerates all the subdelay vectors $T_l$ of $T_l[1, j](1 < j \leq |T_l|)$ and iteratively generates all subvectors of the landmark delay vector combined with the routing vector to expand a landmark path vector into $T_t$. This task is performed iteratively in the planned training set. $[1, j]$ denotes the first router to the jth router sequence. The Algorithm 1 is shown as follows.

---

**Algorithm 1** SubvectorS

---

**Input:**
　　Landmark paths and delay vectors $C_l$
**Output:**
　　Training set $S_n$
1:　$\Theta(T_l[1, j], T_t) \longleftarrow 0, |T_t| \longleftarrow dimension(T_t)$
2:　**for** (int $i = 0$ ; $i < n$; $i$++) **do**
3:　　**for** (int $k = 0$ ; $k < j$; $k$++) **do**
4:　　　$T_{lk} \longleftarrow (T_l[1, k])$
5:　　　$C_{lk} \longleftarrow (R_{cl}, T_{lk})$
6:　　　$S_n.append(C_{lk})$
7:　　**end for**
8:　**end for**
9:　**return** $S_n$

---

### 3.5. Neural Network

The multilayer perceptron consists of multiple perceptrons interconnected with each other, which provides a strong fitting capability. Each perceptron accepts all the outputs from the previous layer. The strength of the connections between the perceptrons is derived from the weights, and an activation function is applied to return the result as the output. A generic example of a two-layer multilayer perceptron is shown in Figure 3, where the first layer is the input layer, Z is the hidden layer neural unit, and y is the output layer.

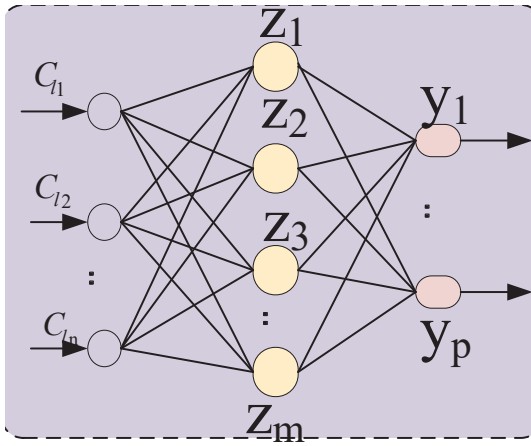

**Figure 3.** MLP general example.

$$Z_m = ReLU(\sum_{i}^{n} W_{im}C_{ln} + b_{im}) \tag{20}$$

$$y_p = softmax(\sum_{j}^{m} W_{mp}Z_m + b_{mp}) \tag{21}$$

$C_{ln}$, $Z_m$, and $y_p$ are the input layer input, hidden layer output, and output layer output of the simple MLP model, respectively. *ReLU* and *softmax* are the activation functions of

the hidden layer and output layer, respectively. $W_{im}$ and $W_{mp}$ are the weight parameters we need to train. $b_{im}$ and $b_{mp}$ is the bias.

### 3.5.1. Model Training

Training: The training of the model is a nonlinear optimization task, which makes the local extremes appear in the task instead of the desired global optimum. Here, we tried to reduce the frequency of the local extremes by various means, firstly by selecting different parameter values during parameter initialization, secondly by applying stochastic gradient descent calculation, and finally, by adjusting the number of training sessions to finally filter the best-performing set of parameters in the training task through repeating the training several times.

Overfitting: The learning performance of backpropagation causes the model to overfit easily. In this paper, the model chooses the L2 regularization to optimize the model training and the overfitting phenomenon. In addition, in the deep learning model, more attention is paid to the model in the validation error rather than the absolute value of the error in the training and validation sets.

### 3.5.2. Network Entity Geolocation

In the target IP localization task, the target IP encoding vector is imported into the trained model, and the result is output through the model to compare the threshold $\tau$. If $Q > \tau$ is satisfied, the localization success is returned, along with the geolocation information (longitude, latitude) of the target IP. Otherwise, the geolocation failure is returned. The Algorithm 2 is shown as follows.

---

**Algorithm 2** SubvectorS_Geo

---

**Input:**
    Target IP path and delay vector $C_t$
**Output:**
    Target IP (longitude, latitude)
1: $C_t \longleftarrow (R_{ct}, T_t)$
2: $Q \longleftarrow MLP(C_t)$
3: if $Q > \tau$
4: return geolocation success,The target IP geographic location is (longitude, latitude)
5: else
6: **return** geolocation failure

---

## 4. Experimental Results and Discussion

Because the previous research on IP geolocation is limited by intellectual property rights and other factors, there is no suitable public dataset available, only the experimental data collected by us.

### 4.1. Dataset

The probes were deployed in Zhengzhou and Hong Kong, China, and Virginia, USA. Traceroute measurements of the acquired IPv6 landmark addresses were performed by probes for a period of a month. The measurement data were collected and combined with methods such as Allys and Mercator [32] for router aliasing and Sarac's [33] anonymous route resolution method for cleaning and filtering the measurement data. We chose the minimum value of delay in the measured data during the long measurement period as the final delay data involved in the experimental training. The hostname data corresponding to the measurement dataset were derived from the public dataset Rapid7 [34]. The IPv6 city-level identification IPs were collected using the method of Zhao [13] for the measured data.

Based on the demand for fine-grained location accuracy in the field of IP geolocation, we screened a total of 5610 street-level landmarks in three regions at home and abroad, including 2195 in Shanghai, China, 1826 in New York State, USA, and 1589 in Tokyo, Japan,

by combining the IPv6 public datasets provided by APNIC [35], an Asian Pacific address distribution organization, and ARIN [36], a North American address distribution organization, through the method of [20]. The region of Shanghai is 6340 square kilometers. The region of New York State is 141,300 square kilometers. The region of Tokyo is 2155 square kilometers. These regions are divided between developed and developing parts of North America and Asia. As shown in Figure 4.

It is worth noting that we were dealing with the cleaned path data. Therefore, the inaccuracy of the route data caused by route aliases and anonymous routers is a problem we must solve before starting the experiment. Routing aliases were cleaned using the Spring [32] approach. As for anonymous routers, we simply treated them as a "top-down" link. The anonymous routing processing scheme of Sarac [33] was used.

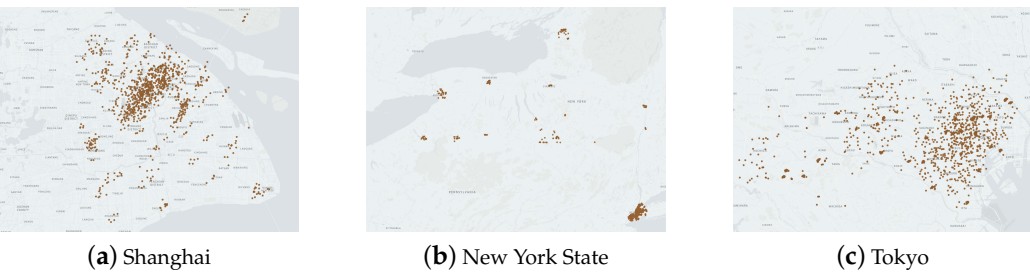

(**a**) Shanghai  (**b**) New York State  (**c**) Tokyo

**Figure 4.** Landmark distribution in the three regions.

### 4.2. Model Parameter Settings

The model implementation was based on the Pytorch framework. The parameters were initialized using the Xavier initialization to select a variety of different parameters. The optimizer was selected as Adam. The L2 regularization term was learned between 0.01, 0.001 and 0.0005. The learning rate was selected between 0.01 and 0.001. The number of hidden layers was selected between 2, 3, 4, and 5. The hidden layer activation function was selected as ReLU, and the output layer activation function was selected as softmax.

### 4.3. Geolocation Experiment Result

Following the settings in Table 1, we collected 5610 landmarks for traceroute measurements in the three regions around the world. We trained the measurements for 60% (3365) of the landmarks in Table 2. Of these 3365 landmark measurements, 20% (1122) were used as the validation set, and another 20% (1123) were selected as the test set. Table 3 shows the geolocation errors that can be geolocated in the three regions for the method proposed in this paper, as well as for the other three IP positioning algorithms. We compared three different IP geolocation algorithms with the geolocation algorithm proposed in this paper under the same landmarks and measurement environments in the three regions. The cumulative distribution of the geolocation errors of the four algorithms for locatable entities is shown in Figure 5.

**Table 1.** Experimental device.

| Device | Region |
| --- | --- |
| Probe deployment | China: Zhengzhou and Hong Kong<br>USA: Virginia |
| Landmark deployment | Shanghai, Tokyo, and New York State |
| Detection protocol | ICMPv6 |

**Table 2.** Dataset details.

| Region | Landmark Deployment | Quantity |
|---|---|---|
| China | Shanghai | 2195 |
| America | New York State | 1826 |
| Japan | Tokyo | 1589 |
| | total | 5610 |

**Table 3.** Performance (kilometers) comparison of baselines and SubvectorS_Geo.

| Method | Shanghai, China | | | New York State, USA | | | Tokyo, Japan | | |
|---|---|---|---|---|---|---|---|---|---|
| | Max | Ave | Med | Max | Ave | Med | Max | Ave | Med |
| IPv6-CBG [1] | 59.892 | 31.802 | 29.254 | 486.729 | 43.603 | 16.105 | 53.766 | 27.587 | 28.088 |
| Corr-SLG [1] | 55.427 | 15.337 | 13.916 | 481.127 | 37.303 | 7.501 | 46.881 | 12.019 | 9.856 |
| TNN [2] | 48.881 | 15.139 | 13.617 | 447.095 | 34.39 | 10.381 | 47.874 | 14.475 | 12.249 |
| MLP-Geo [2] | 46.273 | 13.809 | 11.614 | 428.376 | 27.361 | 8.047 | 43.972 | 12.118 | 10.618 |
| Our Proposed | 44.895 | 11.194 | 9.709 | 421.527 | 24.854 | 7.025 | 42.018 | 10.564 | 8.751 |

"Max" indicates max error distance; "Ave" indicates average error distance; "Med" indicates median error distance. [1] Rule-based IP geolocation algorithm, [2] Deep-learning-based IP geolocation algorithm.

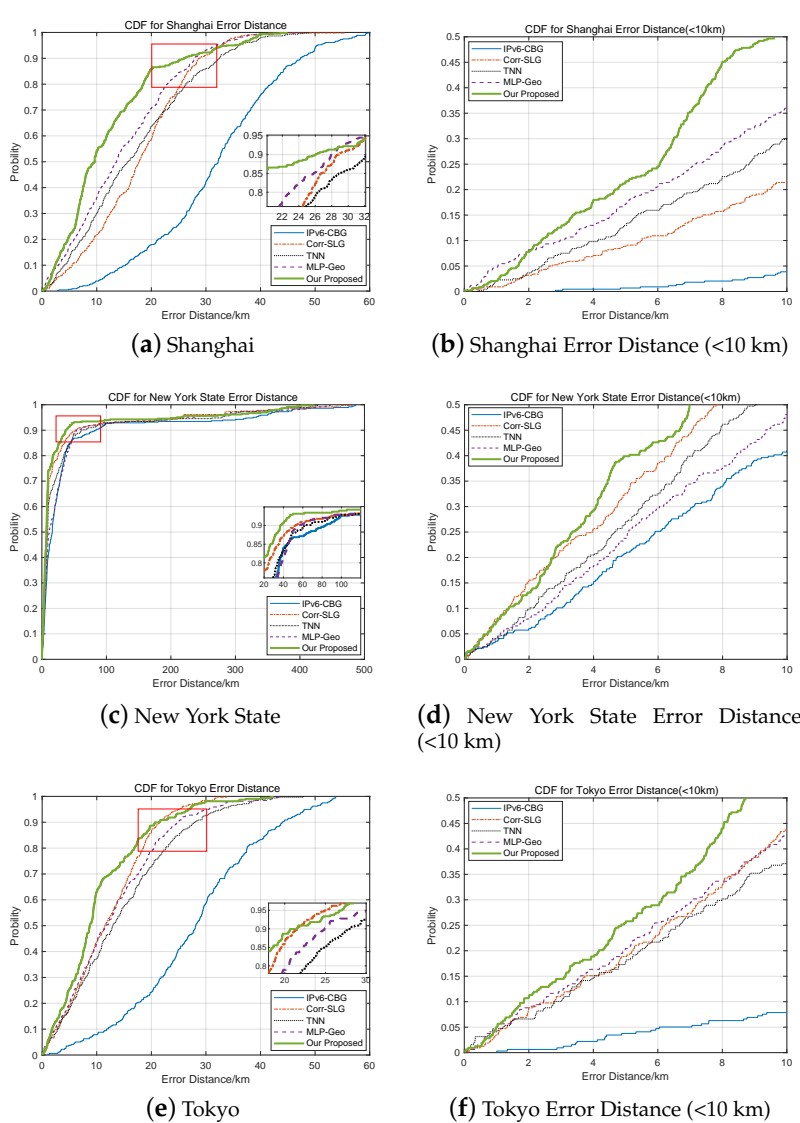

**Figure 5.** Error distance CDF of the three regions.

*4.4. Comparison and Verification*

For accuracy, in Table 3, the median error of the proposed method in the Shanghai dataset was improved by 16.4% compared to the optimal baseline. Compared to the optimal baseline, the maximum error and average error were improved by 3.0% and 18.8%, respectively. In the New York State dataset, the median error, maximum error, and average error values of this algorithm were improved by 6.8%, 1.6%, and 9.2%, respectively, compared to the optimal baseline. In the Tokyo dataset, the median error, maximum error, and average error of this algorithm compared to the optimal baseline were improved by 11.2%, 4.3%, and 12.1%, respectively.

The errors of the four geolocation algorithms in the three different regions were analyzed. Figure 5 shows the cumulative distribution of the geolocation errors for their locatable entities. Furthermore, to show more clearly the geolocation performance of the algorithm at the street level (<10 km), we produced a map of the cumulative distribution of the geolocation errors for locatable entities within 10 km. In addition, the CDF images overlap at large values of the error distances. To solve this problem, we zoomed in on the CDF overlapping region images of each of the three regions. Combined with the overall CDF image in Figure 5, our IPv6 geolocation algorithm outperformed the existing state-of-the-art algorithms for geolocation between 10 km and 50 km.

*4.5. Ablation Study*

In this section, an ablation study was performed to evaluate the effectiveness of SubvectorS in the geolocation task. We named the model with SubvectorS removed and the full model separately as follows. SubvectorS_Geo is the complete model framework. SubvectorS_Geo_A is the model framework for removing SubvectorS. The SubvectorS_Geo_A model replaces SubvectorS with the cosine distance [37]. Furthermore, to increase the effectiveness of SubvectorS in comparison with other similarity measure evaluation methods, the SubvectorS_Geo_B model and the SubvectorS_Geo_C model were added. SubvectorS_Geo_B is a model framework that uses the edit distance on the real sequence (EDR) [38] instead of SubvectorS. SubvectorS_Geo_C is a model framework using the edit distance on real sequence (LCSS) [39] instead of SubvectorS.

As shown in Figure 6, SubvectorS presents a stronger effect on the final IP geolocation accuracy than other similarity metric algorithms. Among them, the cosine similarity has the least impact on IP geolocation accuracy, which is mainly due to the delay vector dimension's nonuniformity of the route hot coding in this paper. Although the EDR and LCSS are less affected by the nonuniformity of the delay vector dimension, the sensitivity of the EDR to noise points leads to a less valid similarity measure than SubvectorS. The LCSS is more consistent with the delay vector of our model, but its minimum distance threshold e is more difficult to define. There is a possibility that the LCSS has the error of returning dissimilar paths as similar results.

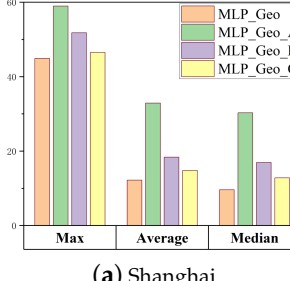
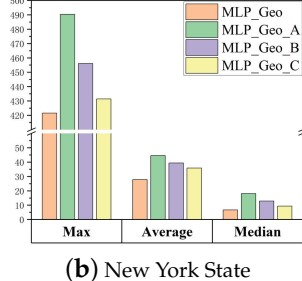
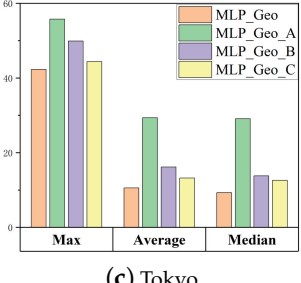

(**a**) Shanghai    (**b**) New York State    (**c**) Tokyo

**Figure 6.** Ablation study in the three experimental regions.

*4.6. Effect of the Number of Hidden Layers on the Model*

In this section, to better optimize and evaluate the geolocation performance of SubvectorS_Geo, we introduced a new evaluation index, the mean squared error (MSE) [40], which

mainly analyzes the influence of the parameters on the mean error distance and reduces the concern about the maximum error distance and median error distance. As shown in Figure 7, on the experimental data of the three real regions, the median error distance and the maximum error distance with the most advantage of the average error distance were always smaller than our selected baseline, which verifies the validity of SubvectorS_Geo. In addition, the best-performing hidden layers in the three experimental regions (Shanghai, New York State, and Tokyo) were 2, 3, and 2, and the worst-performing hidden layers were 5, 1, and 1, respectively. Compared with the worst layers, the improvement of the average error distance was 25.8% on the Shanghai dataset, 23.7% on the New York State dataset, and 22.1% on the Tokyo dataset.

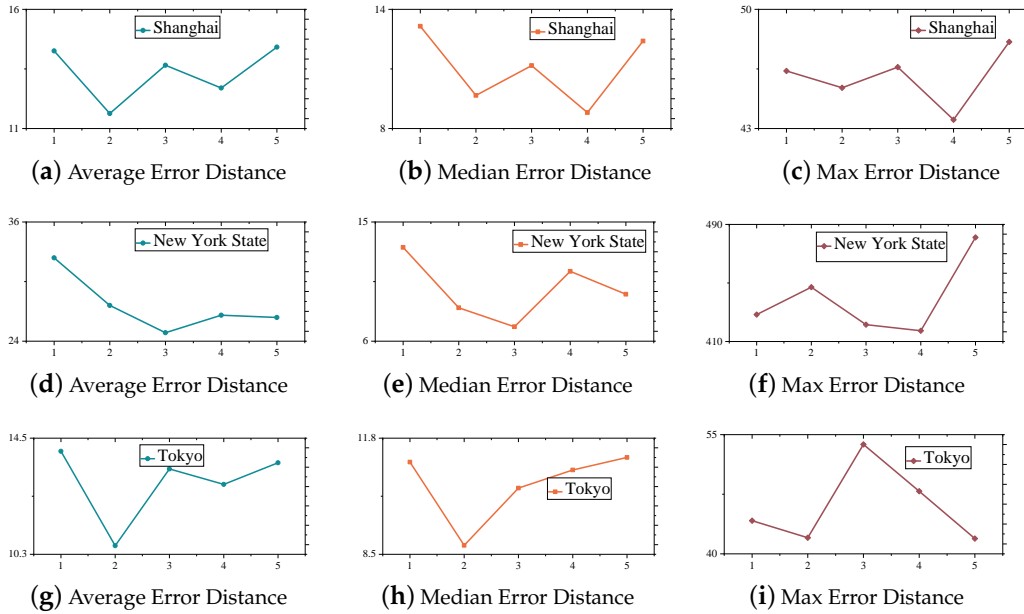

**Figure 7.** Study of hidden layers in the three regions.

### 4.7. Analysis and Discussion

In this section, we analyze the geolocation logic of SubvectorS_Geo, explain it in deep learning, discuss its limitations, and look forward to the fine-grained research in IPv6 geolocation.

SubvectorS_Geo completes fine-grained trusted region partitioning by a layer by layer approximation region constraint policy and, then, evaluates the delayed similarity near the target IP to map the target IP to a landmark that satisfies the threshold. In the region constraint policy design, we introduced AS domains with geographic identification information, identification IPs, closest common routers, and hostname similar routing based on the a priori knowledge of IP geolocation. In addition, IPv6 prefix similarity within the trusted region was also included in the model evaluation, and the assumption that IPv6 addresses in the same region have the same IPv6 address prefix was statistically proven by Padmanabhan [41]. The IPv6 routing lookup algorithm [42] also laterally verifies the validity of this assumption. The network characteristic with geoconstrained information is the logic of our design of the IPv6 geolocation algorithm, which is also the interpretation of SubvectorS_Geo on deep learning. Although SubvectorS_Geo can achieve better geolocation performance in a certain region, it is still in a structured data processing mode and does not focus on the contribution of suboptimal landmarks for the target IP geolocation, in addition to the difficulty of the neural network in modeling the connectivity information of the network topology [43].

Landmarks play an extremely important role in IPv6 geolocation tasks, and the distribution and concentration of landmarks can significantly affect the performance of IP geolocation algorithms [44]. The rapid development of cloud services and the gradual

adoption of the cloud service model for servers in traditional landmark collection regions such as universities, hospitals, and government units, with fewer available high-value landmarks, pose new challenges for IPv6 geolocation technology. It is worth noting that there are more IPv6 application scenarios in the rapidly developing IoT, and IoT devices including cameras are potential landmark collection objects. Meanwhile, we are also committed to obtaining more device network information from IoT application scenarios, and network modeling of IoT IPv6 application scenarios [29] is the direction of our fine-grained IPv6 geolocation research.

## 5. Conclusions

We proposed SubvectorS_Geo, a new IPv6 geolocation algorithm, which screens the routing characteristics of landmark paths based on the assumption of regional delay similarity and applies the idea of layer by layer approximation to divide the hierarchical trusted regions; it provides a new path encoding method that captures the delay vector characteristics near the target IP to achieve high-quality IPv6 geolocation. The final experimental results for three regions showed that the median error distance range of the SubvectorS_Geo geolocation results was 7.025 km to 9.709 km. In IPv6, compared to the current state-of-the-art IP geolocation algorithms, the median error distance was reduced by at least 6.8%. The comparison with the geolocation results of MLP-Geo confirmed the residual path property with a tight delay distribution and strong geographic relationship. We will model the network implementation of IPv6 in graph deep learning for fine-grained IPv6 geolocation algorithms.

**Author Contributions:** Conceptualization, Z.M. and X.H.; methodology, Z.M. and X.H.; software, X.H.; validation, Z.M. and X.H.; formal analysis, X.H. and S.Z.; investigation, X.H. and S.Z.; resources, Z.M.; data curation, X.H. and Q.D.; writing—original draft preparation, X.H.; writing—review and editing, Z.M., X.H. and G.H.; visualization, Q.D.; supervision, Z.M., F.L., Q.Z. and H.W.; project administration, Z.M. and N.L.; funding acquisition, Z.M. and G.H.; All authors have read and agreed to the published version of the manuscript.

**Funding:** This work was supported in part by the Key scientific research project plans of higher education institutions in Henan Province (Grant No. 21zx014), the Guangdong Basic and Applied Basic Research Foundation (No. 2022A1515010417), the Key Project of Shenzhen Municipality (No. JSGG20211029095545002), and the School-enterprise Collab-orative Innovation Project of SZIIT (XQ2021).

**Institutional Review Board Statement:** Not applicable.

**Informed Consent Statement:** Not applicable.

**Data Availability Statement:** Traceroute measurement data for IPv6 can be downloaded here https://github.com/Hxh1863819/SubvectorS_Geo/ (accessed on 4 January 2023).

**Conflicts of Interest:** The authors declare no conflict of interest.

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
