# Peer review of "SubvectorS_Geo: A Neural-Network-Based IPv6 Geolocation Algorithm"

_applsci, doi:10.3390/app13020754_

Round 1
Reviewer 1 Report
The paper proposes a new method to find localization using ipv6. I recommend a sperate section on how this is different from ipv4.
Author Response
尊敬的审稿人:
感谢您对我们题为“SubvectorS_Geo:基于神经网络的 IPv6 地理定位算法”(applsci-2110696) 的手稿发表评论。这些意见都很有价值,对修改和完善我们的论文很有帮助,对我们的研究也有重要的指导意义。我们仔细研究了评论并进行了更正,希望得到批准。修订的部分在纸上用蓝色标记。论文中的主要更正和对审稿人意见的回复如下:
回复审稿人意见:
- Response to comment (Reviewer 1): (The paper proposes a new method to find localization using ipv6. I recommend a separate section on how this is different from ipv4.)
Response: We are very grateful to the reviewers for their recognition and suggestions for adding the study of the difference between the IPv4 geolocation algorithm and the IPv6 geolocation algorithm, which is a very important part of this article because it is the motivation of this article, and according to the suggestions we make the following changes. A comparative study of IPv4 geolocation algorithms and IPv6 geolocation algorithms is added. The design of IP geolocation algorithms is based on IP protocols and the rules of the network environment in which IP protocols are deployed, and we explore the differences between IPv4 geolocation algorithms and IPv6 geolocation algorithms in terms of the differences between IPv4 and IPv6. (Revision: The third paragraph of the introduction, lines 35 to 56.)
我们尽力改进稿件,并对稿件进行了一些修改。这些变化不会影响论文的内容和框架。在这里,我们列出了更改并在修订后的文件中标记了它们。我们非常感谢审稿人的热情工作,并希望更正能够获得批准。再次非常感谢您的意见和建议。
你的
真挚地
马兆瑞

Reviewer 2 Report
Interesting paper deals with real issues, and the results are promising. The main problem with the paper is the misspelling of many words (including section titles) and dividing them by unnecessary hyphens, like dividing between rows of the text in some earlier version of the manuscript. There are no serious issues with grammar - only with spelling, which must be thoroughly corrected.
Paper worth publishing.
Author Response
Dear Reviewers:
Thank you for your comment concerning our manuscript entitled “SubvectorS_Geo: A Neural Network Based IPv6 Geolocation Algorithm”(applsci-2110696). Those comments are all valuable and very helpful for revising and improving our paper, as well as the important guiding significance to our research. We have studied the comments carefully and have made a correction which we hope meets with approval. Revised portions are marked in blue on the paper. The main corrections in the paper and the responses to the reviewer’s comments are as flowing:
Responds to the reviewer’s comments:
1. Response to comment (Reviewer 2): (Interesting paper deals with real issues, and the results are promising. There are no serious issues with grammar - only with spelling, which must be thoroughly corrected.)
Response: Thank you very much for your approval and suggestions on the article, and I will revisit the English writing issues in this article based on your suggestions. We made the following changes. After checking and filtering the words in the text, unnecessary hyphens were removed, and the text was rechecked and revised for word and grammar problems. Word case issues in titles, figure notes, hyphens, and content were modified. For example, in line 198, "net-work" is changed to "network". In line 152, "geo-location" is changed to "geolocation". In line 146, "in-formation" is changed to "information".
We tried our best to improve the manuscript and made some changes in the manuscript. These changes will not influence the content and framework of the paper. And here we list the changes and marked them in the revised paper. We appreciate for Reviewers’ warm work earnestly and hope that the correction will meet with approval. Once again, thank you very much for your comments and suggestions.
Yours
Sincerely
Zhaorui Ma

Reviewer 3 Report
This manuscript proposed a modified IPv6 geolocation method, the topic looks interesting. My comments are as follows:
1) English writing improvement is suggested. For example, the full names of all the abbreviations should be given when they appear for the first time, such as ML and DL in Abstract, or MLP in the caption of Figure 3. Moreover, there are some typos. For example, in line of Abstract, “The classical methods” should be “the classical method”. In line 156 of section 3.1, “net-work” should be “network”.
2) Both motivations and contributions are unclear, please refine them.
3) More comments on the recent advances reviewed in Related Work section are suggested.
4) More evaluation metrics and state-of-the-arts should be included in the experiments to make the experiments more sufficient.
5) Separate discussion section should be considered to discuss both limitations and future directions. Additionally, more IoT applications also are suggested to be discussed or at least mentioned. Some related papers are recommended: HandGest: Hierarchical sensing for robust in-the-air handwriting recognition with commodity wifi devices, IEEE IoT, and online spatiotemporal modeling for robust and lightweight device-free localization in nonstationary environments, IEEE TII.
Author Response
Dear Reviewers:
Thank you for your comment concerning our manuscript entitled “SubvectorS_Geo: A Neural Network Based IPv6 Geolocation Algorithm”(applsci-2110696). Those comments are all valuable and very helpful for revising and improving our paper, as well as the important guiding significance to our research. We have studied the comments carefully and have made a correction which we hope meets with approval. Revised portions are marked in blue on the paper. The main corrections in the paper and the responses to the reviewer’s comments are as flowing:
Responds to the reviewer’s comments:
1. Response to comment (Reviewer 3): (Problems regarding English writing: irregular format of writing acronyms, misspelling of words.)
Response: Many thanks to the reviewers for their advice on English writing, we made the following changes. The ML and DL acronyms in the Abstract have been changed to full names, line 11 "the classical methods" has been changed to "the advanced method", and line 156 of Section 3.1 " net-work" was changed to "network" and is now in section 3.1, line 198. After checking and screening the words in the text, the abbreviation writing format was revised, and the words in the text were rechecked and revised for problems. Case errors in the title, figure notes, hyphenation, and content were corrected.
2. Response to comment (Reviewer 3): (Both motivations and contributions are unclear, please refine them.)
Response: We are very grateful for the suggestions given by the reviewers and we have made the following changes. The contents of motivation and contribution are reworked.
Motivation. In the third paragraph of the introduction section, we discuss the maladaptation of IPv4 geolocation algorithms in IPv6 from a comparative study of IPv4 and IPv6, leading to our motivation. (Revision: third paragraph of the introduction, lines 35 to 56.)
Contribution. In the fourth paragraph of the introduction section, we discuss what challenges we encountered in introducing deep learning in the field of IPv6 geolocation, and our IPv6 geolocation scheme, and refines the contributions as follows. (Revision: Fourth paragraph of the introduction, lines 57 to 84.)
3. Response to comment (Reviewer 3): (More comments on the recent advances reviewed in the Related Work section are suggested. )
Response: We are very grateful for the suggestions given by the reviewers and we have made the following changes. Modifications in the related work section: in the third paragraph of Section 2.1, a measurement-based IP geolocation algorithm is subdivided into a rule-based class and a learning-based class, the latest advances in measurement-based IP geolocation algorithms are cited, and they are analyzed and discussed; in the fourth paragraph of Section 2.1, a discussion of the latest advances in IP geolocation algorithms in IoT-based application scenarios is carried out; in Section 2.2 An analysis and discussion of a rule-based class-based IP geolocation algorithm are added in the first paragraph. (Revision: Section 2.1, third paragraph, lines 109 to 132. Section 2.1, fourth paragraph, lines 133 to 147. Section 2.2, first paragraph, lines 155 to 163.)
4. Response to comment (Reviewer 3): (More evaluation metrics and state-of-the-art should be included in the experiments to make the experiments more sufficient.)
Response: Many thanks to the reviewers for their suggestions on adding experimental indicators, we made the following changes.
Adding new evaluation metrics. In section 4.6, MSE (Mean Square Error) is added, for the study of the number of hidden layers of neural network models, and the mean error distance is evaluated by MSE to verify the validity of the model. (Revision: Add Section 4.6, lines 457 through 468, and add Figure 7.)
Adding state-of-the-art techniques. In sections 4.3 and 4.4, a rule-based IP geolocation algorithm is added as an experimental baseline, Corr-SLG is one of the most advanced techniques in the current IP One of the most advanced techniques in the field of geolocation, the experimental results of Corr-SLG in the current dataset are added and the CDF (cumulative error distribution image) is redrawn in section 4.3, and the analysis and discussion of the experimental results are reworked in section 4.4. (Revision: Add Corr-SLG to Table 3, modify the latest comparison results in section 4.4, lines 417 to 425.)
5. Response to comment (Reviewer 3): ( Separate discussion section should be considered to discuss both limitations and future directions. Additionally, more IoT applications also are suggested to be discussed or at least mentioned.)
Response: Many thanks to the reviewers for their suggestions on the research outlook of IPv6 geolocation in IoT. Based on the reviewers' suggestions, we made the following changes. In the second paragraph of Section 4.7, we add the geolocation logic for SubvectorS_Geo, explain SubvectorS_Geo in deep learning, discuss the limitations of SubvectorS_Geo from the application of neural networks in the field of IPv6 geolocation, and in the third paragraph of Section 4.7, we start from the application scenario of IPv6 and IoT. The research direction of fine-grained IPv6 geolocation is explored. ( Revision: Section 4.7, lines 469 to 498.)
The research on fine-grained IPv6 geolocation in IoT application scenarios is carried out and we get support from the following articles.
Zhang, J.; Li, Y.; Xiong, H.; Dou, D.; Miao, C.; Zhang, D. HandGest: Hierarchical Sensing for Robust in-the-air Handwriting 583 Recognition with Commodity WiFi Devices. IEEE Internet of Things Journal 2022.
Literature Citation Location ([29], Lines 578 to 579.)
Zhang, J.; Li, Y.; Xiao, W.; Zhang, Z. Online Spatiotemporal Modeling for Robust and Lightweight Device-Free Localization in 577 Nonstationary Environments. IEEE Transactions on Industrial Informatics 2022.
Literature Citation Location ([26], Lines 572 to 573.)
Ding, S.; Zhao, F.; Luo, X. A Street-Level IP Geolocation Method Based on Delay-Distance Correlation and Multilayered Common 544 Routers. Security and Communication Networks 2021, 2021.
Literature Citation Location ([10], Lines 539 to 540.)
Li, Q.; Wang, Z.; Tan, D.; Song, J.; Wang, H.; Sun, L.; Liu, J. GeoCAM: An IP-Based Geolocation Service Through Fine-Grained 581 and Stable Webcam Landmarks. IEEE/ACM Transactions on Networking 2021, 29, 1798–1812.
Literature Citation Location ([28], Lines 576 to 577.)
We tried our best to improve the manuscript and made some changes in the manuscript. These changes will not influence the content and framework of the paper. And here we list the changes and marked them in the revised paper. We appreciate for Reviewers’ warm work earnestly and hope that the correction will meet with approval. Once again, thank you very much for your comments and suggestions.
Yours
Sincerely
Zhaorui Ma

Round 2
Reviewer 3 Report
The quality of this paper has been improved, I have no more comments.